# Competing Vegetation Structure Indices for Estimating Spatial Constrains in Carabid Abundance Patterns in Chinese Grasslands Reveal Complex Scale and Habitat Patterns

**DOI:** 10.3390/insects11040249

**Published:** 2020-04-16

**Authors:** Noelline Tsafack, Simone Fattorini, Camila Benavides Frias, Yingzhong Xie, Xinpu Wang, François Rebaudo

**Affiliations:** 1School of Agriculture, Ningxia University, 489 Helanshan West Road, Yinchuan 750021, China; xieyz@nxu.edu.cn (Y.X.); wangxinpu@nxu.edu.cn (X.W.); 2Department of Life, Health and Environmental Sciences, University of L’Aquila, 67100 L’Aquila, Italy; 3Unité Mixte de Recherche (UMR), Evolution Genome Behaviour Ecology (EGCE), French National Research Institute for Development (IRD), French National Centre for Scientific Research (CNRS), Paris-Saclay University, 91190 Gif-sur-Yvette, France; alimac.cherry@gmail.com (C.B.F.); francois.rebaudo@ird.fr (F.R.)

**Keywords:** steppes, landscape ecology, vegetation index, normalized difference vegetation index (NDVI), modified normalized difference water index (MNDWI), Gao’s normalized difference water index (NDWI2), tasseled-cap indices, soil-adjusted total vegetation index (SATVI)

## Abstract

Carabid communities are influenced by landscape features. Chinese steppes are subject to increasing desertification processes that are changing land-cover characteristics with negative impacts on insect communities. Despite those warnings, how land-cover characteristics influence carabid communities in steppe ecosystems remains unknown. The aim of this study is to investigate how landscape characteristics drive carabid abundance in different steppes (desert, typical, and meadow steppes) at different spatial scales. Carabid abundances were estimated using pitfall traps. Various landscape indices were derived from Landsat 8 Operational Land Imager (OLI) images. Indices expressing moisture and productivity were, in general, those with the highest correlations. Different indices capture landscape aspects that influence carabid abundance at different scales, in which the patchiness of desert vegetation plays a major role. Carabid abundance correlations with landscape characteristics rely on the type of grassland, on the vegetation index, and on the scale considered. Proper scales and indices are steppe type-specific, highlighting the need of considering various scales and indices to explain species abundances from remotely sensed data.

## 1. Introduction

Carabid beetle (Coleoptera Carabidae) communities not only depend on the biotic and abiotic features of their local habitat but are also influenced by habitat features at the landscape level [1]. For example, it is known that, in agricultural landscapes, carabid communities are influenced by the composition and structure of the surrounding fields [2,3]. Specifically, vegetation was pointed out as the major landscape factor in carabid ecology because of its pivotal role in providing food, shelter, and overwintering sites [4,5,6,7,8,9]. However, landscape effects can change according to the scale of analysis, and both spatial and temporal scales are long recognized as important parameters in landscape ecology [10,11,12]. Hence, understanding the relationships between land cover and carabid ecology at different spatial scales is an important goal to address conservation actions.

In China, grasslands represent the most widespread ecosystems, accounting for more than 40% of the national land [13], and carabids constitute one of the most abundant beetles in these ecosystems [14,15]. However, Chinese grasslands are under strong anthropogenic pressures due to the effects of climate change and intensive land use [16,17,18,19]. As a result, Chinese grasslands are subject to increasing desertification processes that are profoundly changing land-cover characteristics [20,21,22] with negative impacts on species communities. Due to their prominent role in grassland ecosystems, carabids constitute one of the most investigated animal groups in grassland ecology around the world [23,24,25,26,27,28]. Despite the importance of grasslands in China [29], the influence of landscape characteristics on carabids in Chinese steppes is still completely unexplored.

Several studies argued that carabid abundance is driven by landscape features, especially the amount of vegetation [24,30,31,32]. These studies showed that the normalized difference vegetation index (NDVI) obtained by remote sensing images may be a good predictor of carabid abundance, with positive correlations in both temperate floodplain [32] and open canopy forest landscapes [31]. However, the relationship between NDVI and vegetation can be biased in sparsely vegetated areas, which may make its use problematic when comparing densely vegetated environments with semi-arid, arid, and desert areas (such as in the case of the various steppes considered in the present research). Moreover, NDVI can be useful to model the response of arthropods to plant biomass, but species abundances can be also profoundly influenced by other vegetation-mediated aspects of the environment, such as the amount of moisture, which can be particularly important for carabids [4].

It is known that the response of carabid beetles to landscape characteristics is scale-sensitive [3]; thus, it is important that the influence of landscape characteristics expressed by different vegetation indices is explored at different spatial scales. In this study, we aimed at investigating how landscape characteristics drive carabid abundance in different grassland types (desert, typical, and meadow steppes, according to the classification of Kang et al. [29]) at different spatial scales by using various vegetation indices from Landsat images. In particular, we tested which vegetation indices are best predictors of the observed variance in carabid activity density and at which scale they perform best. Because of the low and scattered vegetation cover in the desert steppe, we expect that the response of carabid activity density to vegetation indices will be maximal at small scales there, where even movements of few meters will lead carabids to abandon favorable conditions. In addition, we expect larger-scale responses in the meadow and typical steppes, which have more uniform conditions, allowing beetles to have longer movements. Our research is innovative in two ways: (1) we used, for the first time, a multiscale approach in carabid landscape ecology based on vegetation indices; (2) we contributed to filling the gap in carabid ecology in grasslands, which are ecosystems of worldwide importance.

## 2. Materials and Methods

### 2.1. Study Area and Carabid Abundances

To investigate the effects of landscape scale and vegetation indices on carabid abundances, we used the dataset of a companion study in northern China grassland ecosystems [33]. Briefly, the study was carried out in three types of steppes in the Ningxia region (northern China, Figure 1), in an area comprised between 36° north (N) and 38° N and between 105° east (E) and 108° E, where three types of grassland ecosystems (desert, typical, and meadow steppes [29]) are present.

Data were gathered from 90 sampling sites placed at random. To reduce spatial autocorrelation problems, sampling sites were separated as much as possible by at least 150 m. Based on the prevailing vegetation, each site was then assigned to one of the three grassland types:

(1) Desert steppe (15 sites), with vegetation mainly represented by drought-tolerant species such as *Agropyron mongolicum*, *Artemisia desertorum*, *A. blepharolepi*, and *Stipa* spp.

(2) Typical steppe (45 sites), with natural patches of grass (*Stipa bungeana*, *S. grandis*, *Artemisia frigida*, *Thymus mongolicus*, *Heteropappus altaicus*, and *Potentilla acaulis*), possibly associated with cut grasses used as fire belts and crops.

(3) Meadow steppe (30 sites), with *Festuca brachyphylla*, *S. bungeana*, *A. frigida*, and *Achnatherum splendens*.

In each site, five pitfall traps were monitored on five dates from May to September 2017. Specifically, pitfall traps were set once a month in mid-month and left in the field for 72 h prior to emptying and removal of the traps. Traps could not operate for more than 72 h, because, in this time span, they became completely full of beetles. Thus, we collected 25 samples (five pitfall traps on five sampling dates) for each site, for a total of 2250 samples, over a period of five months, which allowed us to have a good representation of carabid abundance. This sampling strategy was also used to assure that local beetle populations were not over-sampled.

Pitfall traps consisted of plastic cups (diameter: 7.15 cm, depth: 9 cm) covered by a transparent plastic lid and sunk in the ground, with the cup-lip level with the soil surface. Each trap was filled with 60 mL of a mixture of tap water and vinegar (8%) and sugar (4%) as an attractant, and 70% alcohol (4%) as a preservative. No trap was lost during the sampling. Data from the same site were pooled for the analysis using the sum of carabid catches. A total of 6873 individuals belonging to 25 species were collected. A complete description of the field work including images can be found in Tsafack et al. [33].

### 2.2. Landscape Remotely Sensed Imagery

To compute vegetation indices, we used Landsat 8 Operational Land Imager (OLI) images from the United States Geological Survey (USGS). We extracted the images for which there were no clouds obstructing the sampled points during carabid sampling. As the sample points were located in two different images, we selected two images taken on the same date, i.e., in May 2017 (LC08_L1TP_129034_20170517_20170525_01_T1 and LC08_L1TP_129035_20170517_20170525_01_T1) and in September 2017 (LC08_L1TP_129034_20170906_20170917_01_T1 and LC08_L1TP_129035_20170906_20170917_01_T1). We merged the images for each layer using the “raster” R package [34] to obtain one image in May and one in September. In order to detect and delete small clouds that could not have been identified visually, we used the cloudMask function from the “RStoolbox” R package [35].

### 2.3. Selection of Vegetation Indices

Because different landscape metrics convey complementary information, we considered a wide array of vegetation indices: normalized difference vegetation index (NDVI), tasseled-cap brightness (TC-B), tasseled-cap wetness (TC-W), modified normalized difference water index (MNDWI), soil-adjusted total vegetation index (SATVI), and Gao’s normalized difference water index (NDWI2).

NDVI reflects the photosynthetic activity of vegetation [36] and is particularly useful for evaluating vegetation productivity, health, and structure [31]. Tasseled-Cap indices (brightness and wetness) [37] express more physical features of the landscape, namely, the degree of the landcover brightness and the amount of moisture (wetness) on the soil. MNDWI is an optimized form of the original McFeeters normalized difference water index (NDWI) [38], which eliminates soil and terrestrial vegetation features and retains only open water information in the landscape [39]. NDWI2 [40] is another index useful to depict the landcover related to water features by measuring the liquid water molecules in vegetation. Finally, SATVI [41] was developed to minimize the effect of soil and optimize the vegetation sensitivity accounting for the amount of both green and senescent vegetation.

We computed the tasseled-cap transformation to obtain indices on brightness, greenness, and wetness of ground surface and classical multispectral indices (see Appendix A) using the “RStoolbox” R package [35]. For each sampling site, we extracted the mean value of each multispectral index within a circle with a radius ranging from 25 to 1500 m. This resulted in a matrix of 23 indices at 60 landscape scales for each sampling site.

### 2.4. Sampling Sites

The analyses were performed independently for the different types of steppes to identify the specific effect of vegetation indices in each type of steppe and avoid autocorrelation found in meadow and typical steppes. The autocorrelation was due to the correlations between sampling sites and the altitude gradient in meadow and typical steppes. If the dataset was analyzed globally, an overall clustered distribution of points would affect the results. For each steppe type, we tested for complete spatial randomness of site locations. Spatial randomness was tested with a chi-square test on quadrat counts using the “spatstat” R package [42], which takes the homogeneous Poisson process as the null model for point distribution. For meadow and typical steppes, where sampling sites were split into sub-sampling sectors due to differences in vegetation setting (altitude gradient), we divided the dataset accordingly, resulting in one desert steppe dataset, two meadow steppe datasets, and two typical steppe datasets.

### 2.5. Explaining Carabid Abundance

We used correlation matrices to test how various vegetation indices (Appendix A) were intercorrelated to avoid redundancies in the subsequent analyses. Because NDVI reflects productivity and was proven extremely useful in predicting herbivore and non-herbivore distribution, abundance, and life history traits in space and time for a variety of animal groups [31,32,43,44], we kept the index as a reference and excluded indices that had strong correlations with NDVI (Pearson’s *r* > 0.9).

The sites resulted to be randomly located for the desert steppe, for the two sectors in the meadow steppe, and for the first sector in the typical steppe, but not for the second sector in the typical steppe (see chi-square values in Table 1). Thus, we excluded the four points responsible for non-randomness in the second sector of the typical steppe in all analyses and renamed this ecosystem typical steppe 3 in the rest of the text. We excluded indices that conveyed similar information when they were highly correlated, but we retained indices that were intercorrelated when they had a very different ecological meaning. Thus, we retained the following indices: NDVI, TC brightness and wetness, NDWI2, MNDWI, and SATVI (Table 2).

Selected indices were applied at different spatial scales (a circle with a radius ranging from 25 to 1500 m) to analyze how their variance changed with the scale. Studying the variance as a function of scale is a proxy of spatial heterogeneity [10] and provides information about the vegetation index’s ability at identifying the correct scale to explain carabid abundance. Then, we computed a correlation matrix resulting from the correlation of each vegetation index at each landscape scale with the total abundance of carabids in each type of steppe using the “stats” R package [45]. Finally, we used the highest correlations to identify the indices and the scales that best explained carabid abundance in each grassland type.

Because multispectral indices and scales are correlated, no multiple linear regression models were built using all scales [46]. Instead, we computed the variance inflation factors (VIF) for each index and excluded indices with VIF >10 (which corresponds to a threshold at which the variable is considered to be not independent) using the “usdm” R package [47]. The resulting indices were used to build multiple linear regression models with a stepwise procedure for each grassland type to determine the best model corresponding to the best combination of vegetation indices and the best scale to explain carabid abundances.

## 3. Results

### 3.1. Overall Results

The influence of landscape features on carabid abundance varied according to the type of steppe considered (Table 3). In general, the influence of landscape features on carabid abundance was stronger in the meadow steppe than in the desert or typical steppes. Furthermore, the scale at which the influence was maximal varied according to the type of steppe and the index considered. In general, carabid abundance in the desert and the meadow steppe was correlated with both NDVI and MNDWI, thus reflecting the impact of productivity (NDVI) and water availability (MNDWI), but at different scales. Whereas, in the desert, carabid abundance was correlated with these indices at a short distance (25 m), meadow communities responded at a much larger scale (1250–1500 m). By contrast, carabid beetle abundance in the typical steppe was mostly correlated with the TC wetness at relatively small distance (200 m), with the NDWI2 at an intermediate scale (900 m), and with MNDWI at larger distance (1500 m). Thus, in this ecosystem, carabid abundance responded mostly to wetness at a short distance, to the water in vegetation at the intermediate scale, and to open water at a larger scale. Brightness did not influence carabid abundance in any type of steppe. We also found that the values of vegetation features were higher in autumn than in spring.

### 3.2. NDVI

The abundance of carabids in the desert steppe was negatively correlated with NDVI in spring (May image) at small scales (from 25 m to 100 m). Carabid abundance in the meadow steppe 1 showed more varied responses (negative correlations at scales from 275 m to 375 m, and positive correlations at scales from 750 m to 975 m and from 1125 m to 1500 m) (Figure 2). On this image, NDVI had no effect in the meadow steppe 2 or in the two typical steppe sectors. On the autumn image (September image), only carabid abundance in the meadow steppe showed significant correlations with NDVI. We found varied responses in the meadow steppe 1 (negative correlations at scales 200 m, 225 m, 275 m, and from 525 m to 1150 m, and positive correlations at scales from 1425 m to 1500 m), and in the meadow steppe 2 (negative correlations at scales from 150 m to 575 m and from 750 m to 1500 m). Carabid abundance in the typical steppes was not correlated with NDVI, whether in spring or in autumn.

### 3.3. TC Brightness (TC-B)

In spring, the abundance of carabids in the meadow steppe 2 was significantly correlated with TC-B at large scales (from 1400 m to 1500 m) (Figure 3). In autumn, TC-B was significantly correlated with carabid abundance in the meadow steppe 1 (positive correlations at scales from 600 m to 850 m, and negative correlations at scales from 1350 m to 1500 m) and in the meadow steppe 2 (positive correlations at scales from 50 m to 1500 m). Carabid in the desert and in the typical steppes did not respond to TC-B index, whatever the time (Figure 3).

### 3.4. TC Wetness (TC-W)

In spring, TC-W was significantly correlated with the abundance of carabids in the meadow steppe 2 (negative correlations at scales from 25 m to 100 m, and positive correlations at larger scales, from 1400 m to 1500 m), in the typical steppe 1 (negative correlations at scales from 75 m to 200 m), and in the typical steppe 3 (negative correlations at scales from 175 m to 225 m) (Figure 4). In autumn, TC-W was significantly correlated with carabid abundance in the meadow steppe 1 (negative correlations at scales from 600 m to 850 m), in the meadow steppe 2 (negative correlations at scales 100 m and from 150 m to 425 m, and positive correlations at scales from 825 m to 1150 m), in the typical steppe 1 (positive correlations at scales from 1250 m to 1425 m), and in the typical steppe 3 (negative correlations at scales from 25 m to 50 m, and from 100 m to 225 m). Carabid abundance in the desert did not respond to TC-W index, whatever the time (Figure 4).

### 3.5. NDWI2

In spring, NDWI2 was significantly correlated with the abundance of carabids in the meadow steppe 1 (positive correlations at scales: 150 m to 175 m; 800 m to 950 m; 1175 m to 1300 m and 1400 m to 1500 m), in the meadow steppe 2 (negative correlations at scales from 50 m to 75 m and from 900 m to 1475 m), and in the typical steppe 1 (positive correlations at scales from 600 m to 900 m). In the same time, NDWI2 was not correlated with carabid abundance neither in the desert steppe nor in the typical steppe 3 (Figure 5). In autumn, NDWI2 was significantly correlated with carabid abundance in the desert steppe (positive correlations at scales 25 m and from 175 m to 300 m), in the meadow steppe 1 (negative correlations at scales from 575 m to 950 m), in the meadow steppe 2 (negative correlations at scales from 150 m to 400 m, and positive correlations at scales 700 m, 750 m, 1075 m, 1100 m, 1450 m, and 1500 m), and in the typical steppe 3 (negative correlations at scales 25 m and 50 m) (Figure 5).

### 3.6. MNDWI

In spring, MNDWI was significantly correlated with the abundance of carabids in the meadow steppe 1 (positive correlations at scales from 100 m to 200 m), in the meadow steppe 2 (negative correlations at scales from 50 m to 100 m and from 500 m to 825 m), and in the typical steppe 3 (negative correlations at scales from 25 m to 200 m) (Figure 6). In autumn, MNDWI was significantly correlated with carabid abundance in the desert steppe (positive correlations at scales from 25 m to 100 m and at 175 m), in the meadow steppe 1 (positive correlations at scales from 200 m to 500 m), in the meadow steppe 2 (positive correlations at scales from 275 m to 550 m, and from 750 m to 1400 m), in the typical steppe 1 (positive correlations at scales from 1275 m to 1500 m), and in the typical steppe 3 (negative correlations at scales from 25 m to 250 m) (Figure 6).

### 3.7. SATVI

In spring, SATVI was significantly correlated with the abundance of carabids in the meadow steppe 1 (positive correlations at scales from 850 m to 925 m) and in the meadow steppe 2 (negative correlations at scale 25 m, and positive correlations at scales from 950 m to 1500 m) (Figure 7). In autumn, SATVI was significantly correlated with the abundance of carabids in the meadow steppe 1 (negative correlations at scales from 575 m to 875 m, and positive correlations effect at scales from 1325 m to 1500 m) and in the meadow steppe 2 (negative at scales from 50 m to 1500 m). Carabid in the desert and in typical steppes did not respond to SATVI, whatever the time (Figure 7).

## 4. Discussion

Previous studies on the influence of landscape features on carabid beetles in grassland and agricultural ecosystems were aimed at identifying the effects of spatial heterogeneity expressed by landscape composition (i.e., the variety and abundance of different cover types) and configuration heterogeneity (i.e., the complex spatial arrangement, size, and position of landscape elements or the cumulative length of edges). These studies found that a reduction in spatial heterogeneity decreases carabid abundance [48] and diversity [49], and it impacts negatively on the species with lower dispersal ability, smaller body size [49], and generalist ecology [50].

To the best of our knowledge, however, no study investigated how carabid abundance is influenced by landscape characteristics expressed by vegetation indices from remotely sensed images. This study is, therefore, the first to investigate the spatial patterns of carabid abundance using these indices. Overall, we found that the modified normalized difference water index (MNDWI) was the index with the most significant correlations, thus being the most appropriate index to investigate carabid abundance spatial patterns in the different types of grasslands. The MNDWI was developed to optimize the detection of water features in a landscape [39] and was used to investigate and forecast damages due to agricultural pests, such as aphids [51] and mountain pine beetles [52]. Our study showed that carabids of arid grasslands were also sensitive to this index, thus indicating that water availability is a major driver of carabid abundance.

We predicted that carabid activity density should correlate with vegetation indices at small spatial scale in the desert steppe, due to vegetation patchiness, but at larger scales in the more vegetated steppes. As expected, we found significant correlations at a small scale (300 m) in the desert steppe, and at larger scales (1450 m and 1500 m) in the meadow steppe. Thus, we can assume that mobility of carabids is strongly constrained by vegetation in desert steppe, while carabids species of the meadow steppe are more mobile and can exploit larger habitats [8,33,53]. 

A previous study in European agricultural areas showed that variation in carabid dispersal ability decreases with lower landscape configurational heterogeneity, while higher configurational heterogeneity is associated with lower dispersal ability and smaller carabids [49]. Our findings suggest a different situation, with dispersal reduced in the desert, where there is a high heterogeneity due to vegetation patchiness; however, it is this patchiness of suitable environmental conditions that probably hinders dispersal. Body size of carabids is known to relate to their dispersal ability, with larger species moving longer distances than smaller carabids [49]. However, the desert community is strongly dominated by a large sized species (*Carabus glyptoterus* Fischer Von Waldheim, 1827, 75% of total sampled individuals), which suggests that, in this case, environmental constraints are more important than potential dispersal ability. Although the large-sized *C. vladimirskyi* Dejean, 1830 was dominant in one of the two meadow steppe areas considered (42%), in the other, the small-sized *Poecilus fortipes* Chaudoir, 1850 was dominant (24%). Similarly, the two large species *C. glyptoterus* and *C. vladimirskyi* were dominant in typical steppe 1 (53%) and 2 (33%), respectively. However, in typical steppe 2, the large-sized *Poecilus gebleri* Dejean, 1828 was also dominant (31%). This suggests that links between body size, dispersal ability, and habitat structure are complex, requiring more detailed, species-specific analyses.

Desert carabids also appear to be strongly influenced by productivity, as expressed by the NDVI index, again at a short distance, whereas the same index was correlated with carabid abundance at a larger scale in the meadow steppe. Overall, these findings indicate that desert carabids tend to remain concentrated in the few and scattered vegetation fragments that may provide them with food, shelter, and water.

Water availability is also an important driver for the abundance of carabids in the typical steppe. In this ecosystem, wetness (expressed by TC wetness) influenced carabid abundance at the shorter distance, possibly reflecting the most direct forms of dependence (e.g., drinking, availability of fresh food, reduced transpiration, etc.). The water in vegetation (expressed by the NDWI2) influenced carabid activity at the intermediate scale, thus suggesting that NDWI2 indicates an environmental condition favorable to short movements. Finally, the incidence of open water is important at larger distances, which is consistent with the fact that this parameter measures a large-scale feature of the landscape, suggesting that a prerequisite for carabid abundance may be the presence of distant water and/or their abundance follows some gradient of moisture.

The lack of correlation between NDWI2 and carabid abundance in the meadow steppe suggests that humidity is more homogeneously high, allowing beetles to have activity density patterns independent from this measure. Like carabids, the endangered carrion beetle *Nicrophorus americanus* (Coleoptera Silphidae) is also sensitive to NDWI, and the 800-m scale was the most appropriate scale to investigate its responses to landscape features [54]. The scales of effect in desert and typical steppes obtained with this index confirm the scale of effect found with MNDWI, suggesting that a combination of vegetation patterns and moisture levels influence carabid activity density in these arid environments.

The amount of both green and senescent vegetation (as expressed by the index SATVI) [55] was an important predictor of carabid abundance in the meadow steppe, but a poor predictor in the desert and typical steppes. This result may be due to the higher amount of vegetation cover in the meadow steppe, compared to the other types of steppes.

We found that carabid abundance was not sensitive to the TC brightness index, whatever the type of grassland considered. The TC brightness index is primarily used to map bare soil [56]. The environment in this study is characterized by high bareness of the soil as a result of drought conditions. The lack of effects of TC brightness index may be due to the fact that all vegetation types evaluated are quite open, which makes this index not suitable for this type of ecosystem, at least for carabid landscape pattern studies.

It is important to note that the total percentage of explained variance in our regression was moderate in some cases. This may be due to the fact that we used overall abundances with species that might have idiosyncratic responses. We found the largest proportion of explained variance in the desert, where activity density was dominated by a single species, and the lowest value in typical steppe 2, where two species are equally dominant. It would be interesting, in the future, to separately analyze carabid response to landscape features expressed by vegetation indices for different species, guilds, or functional groups, to explore how differences in species ecology influence activity density patterns. Other community parameters such as richness, diversity, and dominance should be also considered to fully assess grassland carabid community patterns in the landscape context.

## 5. Conclusions

Carabid abundance correlations with landscape characteristics rely on the type of grassland, on the vegetation index, and on the scale considered. In a broader context, such as species conservation or outbreak management, this study shows the need to consider the biology of the species when selecting a vegetation index among those currently available, and that various indices should be taken into account in order to explain variance in species abundances. Regarding appropriate scale for the vegetation indices, results from our study indicates that different scales should be considered depending on landscape type. For landscape ecology studies that make use of remotely sensed data, this calls for a more systematic consideration of the different indices available at different spatial scales, a practice that should become more widespread.

## Figures and Tables

**Figure 1 insects-11-00249-f001:**
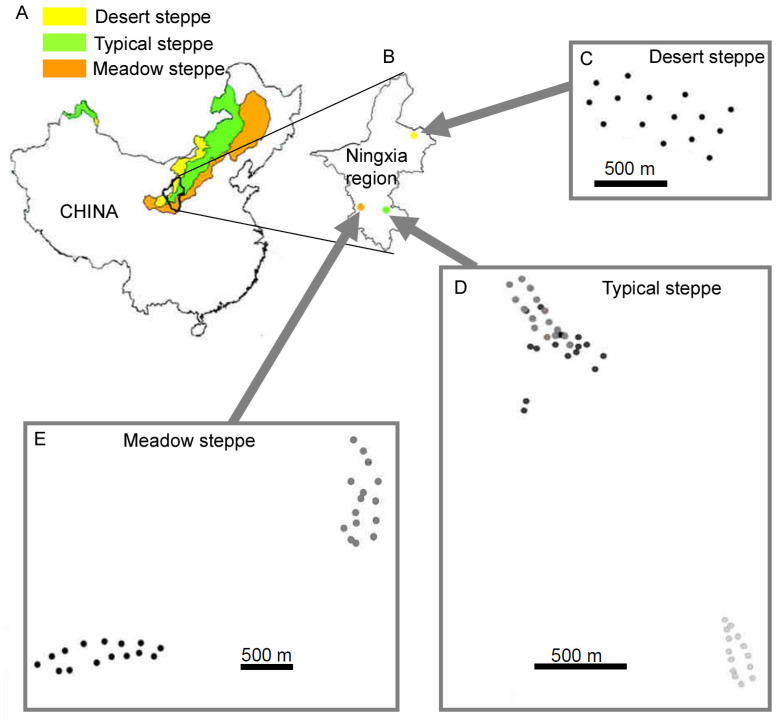
Study area and sampling sites: (**A**) distribution in China of the three grassland types investigated in the present research (redrawn from Kang et al. [29]); (**B**) location of the three study areas, representative of the three grassland types, in the Ningxia region; (**C**–**E**) distribution of sampling sites. Dots of different colors within the same panel indicate that they belong to different sectors (identified on vegetation). Due to relief, some sampling sites may appear closer than they were in the reality.

**Figure 2 insects-11-00249-f002:**
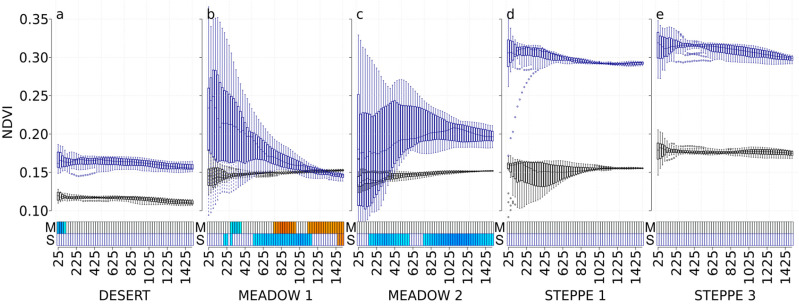
NDVI values at each scale and correlations with carabid abundances for the desert steppe (**a**), the meadow steppe 1 and 2 ((**b**,**c**), respectively), and the typical steppe 1 and 3 ((**d**,**e**), respectively). Boxplots represent the NDVI values with a black border for the May 2017 Landsat image and with a dark-blue border for the September 2017 Landsat image. For each boxplot, a different background color indicates a significant difference in NDVI values using the Tukey test. The *x*-axis presents the different spatial scales used in the analysis (cardinal distances in meters representing circle radius ranging from 25 to 1500 m). The correlation with carabid abundances is represented with rectangles at the bottom of each graph, using a gradient from blue to red (dark blue corresponds to a negative correlation, dark red corresponds to a positive correlation, and white corresponds to no significant correlation).

**Figure 3 insects-11-00249-f003:**
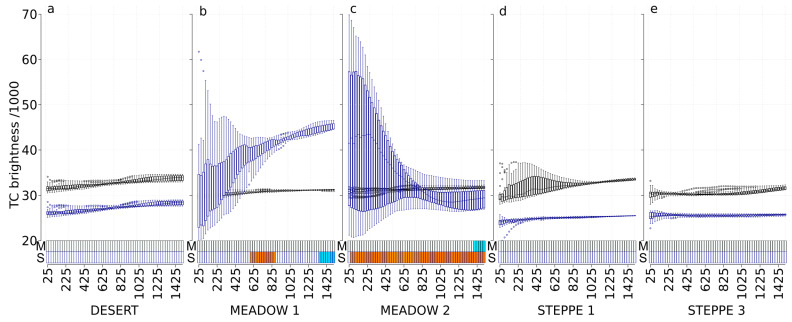
Tasseled-cap brightness values at each scale and correlations with carabid abundances for the desert steppe (**a**), the meadow steppe 1 and 2 ((**b**,**c**), respectively), and the typical steppe 1 and 3 ((**d**,**e**), respectively). Boxplots represent the tasseled-cap brightness values with a black border for the May 2017 Landsat image (M) and with a dark-blue border for the September 2017 Landsat image (S). For each boxplot, a different background color indicates a significant difference in tasseled-cap brightness values using the Tukey test. The *x*-axis presents the different spatial scales used in the analysis (cardinal distances in meters representing circle radius ranging from 25 to 1500 m). The correlation with carabid abundances is represented with rectangles at the bottom of each graph, using a gradient from blue to red (dark blue corresponds to a negative correlation, dark red corresponds to a positive correlation, and white corresponds to no significant correlation).

**Figure 4 insects-11-00249-f004:**
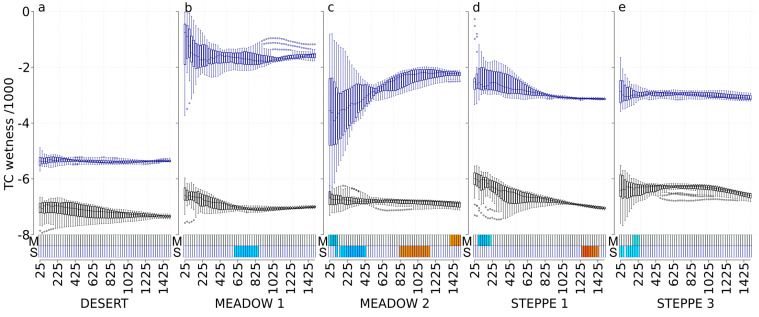
Tasseled-cap wetness values at each scale and correlations with carabid abundances for the desert steppe (**a**), the meadow steppe 1 and 2 ((**b**,**c**), respectively), and the typical steppe 1 and 3 ((**d**,**e**), respectively). Boxplots represent the tasseled-cap wetness values with a black border for the May 2017 Landsat image (M) and with a dark-blue border for the September 2017 Landsat image (S). For each boxplot, a different background color indicates a significant difference in tasseled-cap wetness values using the Tukey test. The *x*-axis presents the different spatial scales used in the analysis (cardinal distances in meters representing circle radius ranging from 25 to 1500 m). The correlation with carabid abundances is represented with rectangles at the bottom of each graph, using a gradient from blue to red (dark blue corresponds to a negative correlation, dark red corresponds to a positive correlation, and white corresponds to no significant correlation).

**Figure 5 insects-11-00249-f005:**
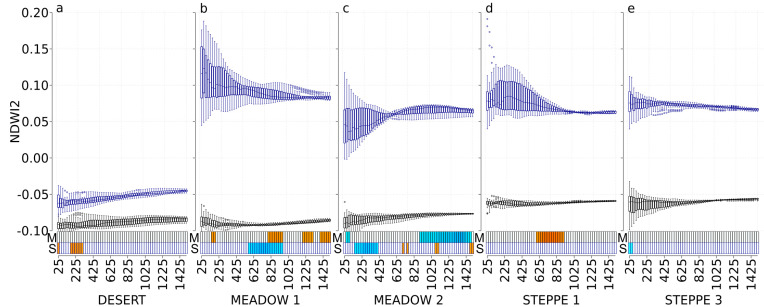
NDWI2 values at each scale and correlations with carabid abundances for the desert steppe (**a**), the meadow steppe 1 and 2 ((**b**,**c**), respectively), and the typical steppe 1 and 3 ((**d**,**e**), respectively). Boxplots represent the NDWI2 values with a black border for the May 2017 Landsat image (M) and with a dark-blue border for the September 2017 Landsat image (S). For each boxplot, a different background color indicates a significant difference in NDWI2 values using the Tukey test. The *x*-axis presents the different spatial scales used in the analysis (cardinal distances in meters representing circle radius ranging from 25 to 1500 m). The correlation with carabid abundances is represented with rectangles at the bottom of each graph, using a gradient from blue to red (dark blue corresponds to a negative correlation, dark red corresponds to a positive correlation, and white corresponds to no significant correlation).

**Figure 6 insects-11-00249-f006:**
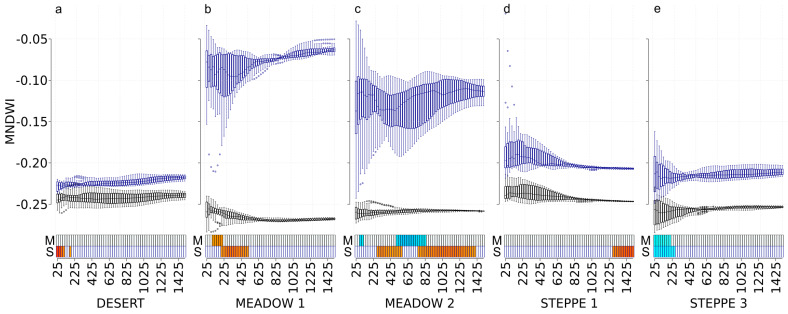
MNDWI values at each scale and correlations with carabid abundances for the desert steppe (**a**), the meadow steppe 1 and 2 ((**b**,**c**), respectively), and the typical steppe 1 and 3 ((**d**,**e**), respectively). Boxplots represent the MNDWI values with a black border for the May 2017 Landsat image (M) and with a dark-blue border for the September 2017 Landsat image (S). For each boxplot, a different background color indicates a significant difference in MNDWI values using the Tukey test. The *x*-axis presents the different spatial scales used in the analysis (cardinal distances in meters representing circle radius ranging from 25 to 1500 m). The correlation with carabid abundances is represented with rectangles at the bottom of each graph, using a gradient from blue to red (dark blue corresponds to a negative correlation, dark red corresponds to a positive correlation, and white corresponds to no significant correlation).

**Figure 7 insects-11-00249-f007:**
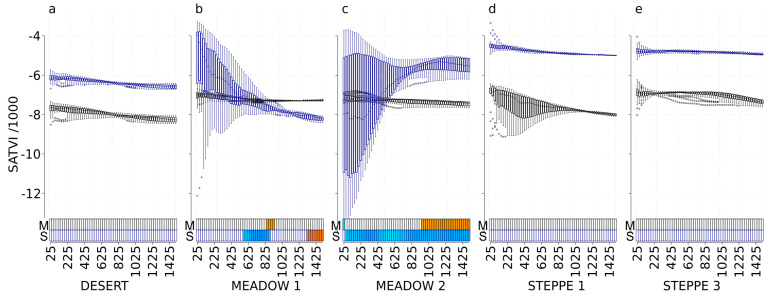
SATVI values at each scale and correlations with carabid abundances for the desert steppe (**a**), the meadow steppe 1 and 2 ((**b**,**c**), respectively), and the typical steppe 1 and 3 ((**d**,**e**), respectively). Boxplots represent the SATVI values with a black border for the May 2017 Landsat image (M) and with a dark-blue border for the September 2017 Landsat image (S). For each boxplot, a different background color indicates a significant difference in SATVI values using the Tukey test. The *x*-axis presents the different spatial scales used in the analysis (cardinal distances in meters representing circle radius ranging from 25 to 1500 m). The correlation with carabid abundances is represented with rectangles at the bottom of each graph, using a gradient from blue to red (dark blue corresponds to a negative correlation, dark red corresponds to a positive correlation, and white corresponds to no significant correlation).

**Table 1 insects-11-00249-t001:** Chi-square tests based on quadrat counts for testing complete spatial randomness of site locations; df—degrees of freedom.

Site	χ^2^	df	*p*-Value
Desert steppe	16.667	24	0.2749
Meadow steppe 1	26.667	24	0.6405
Meadow steppe 2	20	24	0.6064
Typical steppe 1	16.667	24	0.2749
Typical steppe 2	53.333	24	0.0011
Typical steppe 3 (Typical steppe 2 with 4 points excluded)	24	24	0.9232

**Table 2 insects-11-00249-t002:** Correlation between vegetation indices for the two sampling periods: May (below the diagonal) and September 2017 (above the diagonal). Brightness: tasseled-cap brightness, wetness: tasseled-cap wetness, MNDWI: modified normalized difference water index, NDVI: normalized difference vegetation index, NDWI2: Gao’s normalized difference water index, SATVI: soil-adjusted total vegetation index.

May/September	Brightness	Wetness	MNDWI	NDVI	NDWI2	SATVI
Brightness	−	0.53 ∙	0.88 ***	−0.70 **	0.26	−0.91 ***
Wetness	−0.72 **	−	0.74 *	0.09	0.93 *	−0.13
MNDWI	0.70 **	−0.07	−	−0.60 *	0.47	−0.70 **
NDVI	−0.80 **	0.90 ***	−0.36 *	−	0.42	0.88 ***
NDWI2	−0.29	0.83 *	0.34	0.75 *	−	0.16
SATVI	−0.98 ***	0.81 **	−0.63 *	0.87 ***	0.40 ∙	−

Significance codes: *** *p* = 0.001, ** *p* = 0.01, * *p* = 0.05, ∙ *p* = 0.1.

**Table 3 insects-11-00249-t003:** Multiple linear regression models of carabid abundances with multispectral indices after excluding vegetation indices with collinearity problem and a stepwise procedure to select the best model on the remaining vegetation indices (NDVI, tasseled-cap brightness, tasseled-cap wetness, NDWI2, MNDWI, and SATVI). Spatial scales (cardinal distances in meters representing circle radius ranging from 25 to 1500 m) at which the effects of estimators of indices ((+) for positive and (−) for negative)) explaining carabid abundance are significant. Models with tasseled-cap brightness were never better than the others; however, when this variable was considered alone, it was significant at some scales in Meadow 1 and Meadow 2.

	Desert		Meadow 1		Meadow 2		Typ. Steppe 1		Typ. Steppe 3	
	May	Sep	May	Sep	May	Sep	May	Sep	May	Sep
NDVI	(−)25 *					(−)1250 **				
Brightness										
Wetness							(−)200 *			
NDWI2		(+)300 *					(+)900 *			(−)25 ∙
MNDWI		(+)25 ***	(+)1500 *		(+)1450 *			(+)1500 **	(−)25 **	(−)250 ∙
SATVI				(−)750 **						
F	8.406	20.92	5.874	11.59	5.663	9.57	6.945	12.95	7.845	4.747
df	1/13	2/12	1/13	1/13	1/13	1/13	2/12	1/13	1/24	2/23
*p*-value	0.012	<0.001	0.031	0.005	0.033	0.009	0.010	0.003	0.010	0.019
*r*-squared	0.393	0.777	0.311	0.471	0.303	0.424	0.537	0.499	0.246	0.292

Significance codes: *** *p* = 0.001, ** *p* = 0.01, * *p* = 0.05, ∙ *p* = 0.1.

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
