# Peer review of "Competing Vegetation Structure Indices for Estimating Spatial Constrains in Carabid Abundance Patterns in Chinese Grasslands Reveal Complex Scale and Habitat Patterns"

_insects, 2020, doi:10.3390/insects11040249_

Round 1
Reviewer 1 Report
General comments
The manuscript (ms) submitted as an “article” to the journal Insects. The scopes and the suggested outline of the ms suits well with the journal’s mission and I’m sure that the ms will attract many readers to the journal. The ms generally well-written the language and style are decent; in some points, statements are rather short, need some further details to be clear for readers. In addition in some cases there some strange and misty statements in the text, thus some revision by a native speaker may improve the overall quality of the ms. The suggested analyses are well built, suits well to the concept of the paper. There are some conceptual flaws in the ms,which can be solved before publication: 1.) the paper focus only the carabids abundance data, however the species richness and diversity can be also interesting for consideration. 2.) the organization of the results is not well-established, since it follows the steps of the analyses and not the concept/scope of the ms, 3.) the discussion part do not include any real literature overview, however it would be great to see how the authors connect their findings with the available evidences (references). 4.) the organization of the figures might be improved, since in some points there are missing details which makes the readability difficult. All of these issues will be discussed in details below in the section specific comments.
Based on all of these major issues raised above I think that the authors made a neat manuscript which is ready for publication after a concise revision.
Specific comments
Lines 2-4: In my opinion the title is a bit clumsy, it simply tells, the scope of the paper, however the a question style title, or a pattern description would be better, here is my treat: Travelling without moving: competing vegetation structure indices for estimating spatial constrains in carabids abundance patterns in Chinese grasslands.
Line 11: In affiliation #3, there are some missing details, such as postal code, in addition please do not abbreviate the name of the university, and please use English name for institutions.
Lines 22-23: This sentence is not so clear, my reading is suggest that MNDWI index was the best one for the analyses. If so, that is this not a results only an intermediate step during the analyses toward the real results.
Lines 25-26: The same as in the previous comments. In general the abstract includes mostly methods, no real results given, thus we do not know, how the index can explain carabids abundance in steppes. In addition there is no details given about which type of the grasslands were studied in which context.
Lines 71-72: narrow scale?, you mean that short distances?
Line 73: should write: In addition, we expect….
Lines 77-78: I appreciate the enthusiasm, but why the Chinese grassland are so important on a global scale? Some further clarification needed, or rephrase the sentence.
Lines 86-87: If you analyzed separately sites with at lest 150m distance, that means that you have a spatial autocorrelation, since if you do not have any, then you would not implement separate analyses. I believe the potential reader will figure out on the same way, so please justify better with references.
Lines 100-103: You mention here only carabids abundance, but what is about the species richness?
Lines 132-152: This part is not about the index selection, but is more about the description of some indices (but not the complete list which is implemented in the ms.), thus this part is largely overlap with Table 1. I don’t think that is necessary to describe all index, instead you focus on the grouping of the indices based on their nature.
Lines 154-155: It is so sad that you analyzed the different steppes separately, since the major novelty of the study has lost. There is no overall comparison between sites/steppe types and different spatial scales.
Line 169: Table 1 supposed to be move to Appendix, since it just describe the indices.
Line 188: You mention here “different spatial scales”, but there is no any definition for which range of spatial scales (ie. distances between 25-5600m) considered during the analyses.
Line 202: One of the major issues with the results, that the presentation is based on the vegetation indices, however the primary purpose of the paper to explore that how the different steppe types and spatial distances are related with the carabids’ abundance. It is so difficult to read out these patterns in the current form. In addition, another issue is that the description is quite mechanistic, so the result describe what we can see on the figures and tables already, but not biological context presented. Let’s give and independent and random example: mechanistic presentation: the length of the elytra was 6.2 and 2.2 mm respectively for group A and B. Results with biological context: the females (group A) of xy species were larger than males (B) based on the body size estimation by elytra length (6.2 and 2.2 mm respectively). I hope that it help to explore the issue with the results.
Line 223: Table 3: for the indices in the table are the cardinal distances are given which were significantly correlated with carabid abundance? If so please explain in the legend.
Line 237: Figure 3, but for all figures, please specify that the x axis is about the cardinal distances in meters. By the way, it is quite evident that there are cardinal distances were given during in the analyses, but not mentioned in the text.
Lines 341-344: The discussion part is one of the least appropriate parts of the ms. There is no real structure just simply present the overview of the results to compare what we already know abut the major vegetation indices. I still missing the context for sites/steppe types and different spatial scales for carabids. In addition there are many excellent paper published about the spatial ecology for carabids in from grasslands (please see: Tscharntke, Batáry P and Gallé R publications), but this feature/context have not discussed.
Author Response
Dear Reviewer,
Thank you very much for giving us the possibility of submitting an improved version of our manuscript submitted to the journal Insects.
We are very grateful for your very detailed and useful comments.
We incorporated all corrections and suggestions.
Please see the attachment.
All the best,
Noelline Tsafack
(on behalf of all coauthors).

Reviewer 2 Report
This is a well designed study testing models for three different grassland types to be able to calculate carabid abundance. The best thing about this paper is that these models are tested against actual data from a previous paper, and show what are the best characteristics to be used if this is applied to other grassland areas in the world. This will be very useful for designing many future projects.
The statistics are laid out clearly, and are easily followed. However, I am not a fan of Figures 3 to 8, where you illustrate your results. As they currently stand, they are too small, making them difficult to interpret. This slowed me down while reading, and caused more confusion than helped, which indicates that a change is needed.
First, I would make them bigger. This would solve a lot of the issues. You all clearly spent some time on them, so I would increase their size. Maybe have them landscape on the page?
Second, what do the individual box plot (?) colors mean? A good example is the rainbow color pattern in Figure 3b for the September 2017 values. I have a question mark near the box plot above, as the only way I can guess these are box plots is due to in-text clues. I couldn't find anything about what the colors mean, so if it is in the text, it should be clearer. I know it relates to significance, but what is yellow relative to blue? I also don't think this really helps either, and makes it more distracting than anything else. If things could be removed, I would remove the individual coloring of box plots.
Finally, it took me quite a bit to realize that the bottom rectangles were blue. This is an issue with the small size of the figures. I would add a 'M' and a 'S' to the left most side of each row, as it would be another indicator that the top row of rectangles (black) refers to May data, and the bottom (blue) refers to September. This would also help in case someone printed this out in black and white to read.
If you can't make the figures bigger, I would recommend these things as essential. Get rid of color in the individual box plots, keep it blue or black as appropriate. These colors just confuse and don't help. What the reader cares about is already in the rectangles below. Add 'M' and 'S' to y-axis of rectangles of Fig. Xa, as it adds a layer of redundancy to the figure, and doesn't rely on the color of the rectangles. Also Figure 7 is missing a-e.
As for spelling and grammar, everything read easily and I noticed no glaring spelling or grammar issues. The one issue I noticed was line 380 with the sentence ending "... quite open t." This looks like a clipped sentence due to previous rounds of editing to me. I would like to see this conclusion finished, as it reads as if something is missing.
I think if you follow my recommendations about the figures, you all will have a great manuscript and I look forward to seeing it in print.
Author Response

(The authors gave the same response as above.)

Reviewer 3 Report
Title: Landscape drivers of carabid abundance in northern Chinese grasslands: Vegetation indices and scale effects
Dear Editor and Authors,
The authors investigated how landscape characteristics drive carabid abundance in different steppes at different spatial. For this, they used pitfall traps to capture carabid beetles and satellite images to visualize vegetation characteristics; so, they intend to improve methods to infer the best predictors of the distribution and abundance of this group of predator beetle. The authors used several complementary vegetation indices to test the hypothesis, as well as performing robust and adequate statistical analyzes. The work is excellent and innovative and will undoubtedly help to test many ecological hypotheses in areas with similar vegetation on Earth, such as the African and South American savannas. The authors themselves express the main criticism of the work at the end of the discussion: the biology of different species of Carabidae is not present in work. At least the behaviour of dominant species should be mentioned and discussed.
Minor points
L41 - “patial” - spatial?
L91- “S. grandis ”in italic.
Author Response

(The authors gave the same response as above.)

Reviewer 4 Report
Introduction: the aim of the study should be at the end of the introduction, it should be the consequence of the critical points raised in the itroduction
41 change "patial to "spatial"
183 fig.2 is fine but it should be more clear to have numbers instead of colored quadrants
209 it seems quite difficult that an index exerts an influence on carabids, probably it is better to change the use of the word "influence" (and derivates) with "correlation"
223 table3 I have some perplexity about the r-squared values, because they give the proportion of variability explained by your correlation, so I see many low values, e.g. NDVI in Desert May r-squared=0.392 means that near 40% of variation is explained by your correlation while the remaining 60% is unknown, moreover your result is not by chance (i.e. significance =0.05), so I'm afraid that such disproportion (40% vs 60%) is a little bit not confirming your line.
238 fig3 is quite difficult to read, it is too small. In the legend write that the x axis units is distance from the sample site centre
The same for figgs 4 to 8
344 the MNDWI shows low r-squared values in table3, so I suspect that correlation is not so high.
354 in your study you didn't mention any single species, there are no species ecology description, so to assume mobility features about uknown species is more than your data suggest, because it is not possible to have an idea of the consequences of the dispersal power of the real species
359 your result supports (within the limit of r-squared values) that MNDWI possibly is correlated with carabid abundance, then you argue that carabids are sensitive to moisture, but again without knowing anyting about the species that you have collected
395 I would be more cautious, I'm afraid that given the above, your study doesn't show the need to consider the biology, instead it shows that your results should be validated on the light of species biology
Author Response

(The authors gave the same response as above.)

Round 2
Reviewer 1 Report
The authors have made a decent revision. I have no any further suggestion and glad to suggest for accepting the manuscript as it is.
Author Response
Dear Reviewer,
Thank you for appreciating our revision.
All the best,
Noelline Tsafack
(on behalf of all coauthors).
Reviewer 4 Report
52-54 here the aim of the study is given, then the expected results are given at the end of the introduction. Please join aim with expected results
178 Figure 2 again, it is a beautiful picture but it is not good for scientific information, because with circles the quantitative information is proportional to the angle of the sector, so how much is the negative correlation between wetness and brightness? a little bit less than 3/4 of -1?
Then, the bar at the bottom of the half matrices is a legend, while it it could be confused as part of the pictured data, moreover it is organized as shades of different colors that the reader has to compare with those in the circles.
This is not a picture for a scientific paper.
The same (even if not worst) for supplmentary materials
204 "strongly correlated": the authors seem to ignore the meaning of correlation, or they confuse it with that of significance
r-squared is the coefficient of determination, whose squared root gives the correlation coefficient
the squared roots of the sequence in table 2 give
0.63 0.88 0.56 0.69 0.55 0.65 0.73 0.71 0.50 0.54
where I can see only one (0.88) high correlation (not strong)
so please avoid "strongly" at least
Throughout the text it remains obscure why low correlations are so important
Table2 is lacking in information, because there is no way to know the kind of correlation (positive or negative), that is given only after deciphering figgs 3 to 8
Figures 3-8 are very very difficult to read, I have to zoom in every single sub-picture. The background color of the boxplots sometimes is the same as for the correlation in the rectangles at the bottom, where the blue border of the (small) rectangles gives a faint of blue to the rectangles area.
In these pictures blue is used for negative correlation, i.e. it is inconsistent with fig2 where it was used for positive correlation.
If you want to convey information with colors, clear and well contrasted pictures should be drawn, otherwise I have simply to trust the color carousel given by the authors.
233 sorry, here (and hereinafter) I'm lost. About TC-B I don't see any significative value in Table 2
336 no, MNDWI was the index with the most significant correlations, even if the correlations were not so high
376 are you suggesting that carabids can intentionally look for (distant) water?
Author Response
Dear Reviewer,
Thank you very much for giving us the possibility of submitting an improved version of our manuscript submitted to the journal Insects.
We are very grateful for your comments.
We incorporated all corrections and suggestions.
Please see the attachment.
All the best,
Noelline Tsafack
(on behalf of all coauthors).
